# Teacher Forcing Recovers Reward Functions for Text Generation

**Yongchang Hao**[†]**, Yuxin Liu**[†]**, Lili Mou**[†‡]
[†]Dept. Computing Science, Alberta Machine Intelligence Institute (Amii)
University of Alberta, Canada
[‡]Canada CIFAR AI Chair, Amii
{yongcha1,yliu17}@ualberta.ca, doublepower.mou@gmail.com

## Abstract

Reinforcement learning (RL) has been widely used in text generation to alleviate the exposure bias issue or to utilize non-parallel datasets. The reward function plays an important role in making RL training successful. However, previous reward functions are typically task-specific and sparse, restricting the use of RL. In our work, we propose a task-agnostic approach that derives a step-wise reward function directly from a model trained with teacher forcing. We additionally propose a simple modification to stabilize the RL training on non-parallel datasets with our induced reward function. Empirical results show that our method outperforms self-training and reward regression methods on several text generation tasks, confirming the effectiveness of our reward function.[1]

## 1 Introduction

Teacher forcing [7] is the common training method for text generation models. Although this practice has been widely applied [7, 13, 58], there are two main issues: 1) Teacher-forcing training is data-hungry because parallel datasets are usually expensive to obtain. On the other hand, there are numerous unlabeled, non-parallel datasets available. This poses an urge to efficiently exploit non-parallel data. 2) Teacher forcing introduces a discrepancy between training and inference because the model learns to predict the next word based on the partial groundtruth reference during training, whereas in inference the model predicts the next word based on its self-generated previous words. This undesired discrepancy is known as *exposure bias* [44, 6, 28, 59].

To address the first problem, a straightforward method is to generate pseudo-parallel sentences for data augmentation, such as self-training [2], sequence-level knowledge distillation [23], and back-translation [49]. However, the exposure bias remains in such cases.

To address the second problem, the model should be trained on self-generated sentences. Common solutions are often based on reinforcement learning (RL). In text generation, however, there does not exist a naturally defined reward function for RL. Researchers have proposed various heuristic scores as the reward, such as BLEU [38] for translation and ROUGE [31] for summarization. These reward functions are task-specific and not generalizable to other tasks. Further, these rewards require parallel data, failing to address the first problem above; they are typically sparse (only non-zero at the end of a sentence), making RL training difficult.

The goal of this paper is to address these two problems in one framework with a learned, dense reward function. Our approach has two steps: we first train a sequence-to-sequence (seq2seq) model on the parallel dataset and induce a reward function from the model. Then, we apply RL on non-parallel data based on our induced reward function.

---

[1]Our code is publicly available at https://github.com/MANGA-UOFA/LMReward

36th Conference on Neural Information Processing Systems (NeurIPS 2022).

Our method is task-agnostic and does not require handcrafted engineering or heuristics. Further, our reward function provides dense (step-wise) training signals, which makes RL training much easier than sparse rewards. Additionally, the reward function derived from the seq2seq model does not directly participate in the generation, which allows the model to explore based on its own prediction and thus alleviates the exposure bias.

We conduct experiments on dialogue generation and paraphrase generation. The empirical results suggest that our method leads to better performance compared with several baselines, including self-training and task-specific heuristic reward learning, on both tasks. This confirms the effectiveness and generality of our framework.

## 2 Approach

Our approach trains the seq2seq model on non-parallel data with reinforcement learning, whose foundation is the Markov decision process (MDP). In this section, we first introduce the MDP formulation for text generation. Then we describe our method to derive the reward function from a seq2seq model trained by teacher forcing. Finally, we describe the policy gradient method used in RL training with our induced reward function.

### 2.1 Reinforcement Learning Formulation of Text Generation

**Text Generation as a Markov Decision Process (MDP).** We formulate the text generation process as an (undiscounted) MDP, which can be represented as a tuple $(\mathcal{S}, \mathcal{A}, T, r)$. At every step, a decision $a \in \mathcal{A}$ is made based on its state $s \in \mathcal{S}$. The transition dynamic $T(s'|s, a)$ is the probability of the next state being $s'$, given the current state $s$ and the action $a$. A function $r : \mathcal{S} \times \mathcal{A} \to \mathbb{R}$ defines the reward based on a state and an action.

Typically, the decision making is assisted by a policy $\pi$, which is a predicted distribution over actions and is trained to maximize the expected total reward, also known as an action value function:

$$q^{\pi}(s, a) := \mathop{\mathbb{E}}_{\substack{a_t \sim \pi(\cdot|s_t) \\ s_{t+1} \sim T(\cdot|s_t, a_t)}} \left[ \sum_{t=1}^{H} r(s_t, a_t) | s_1 = s, a_1 = a \right], \tag{1}$$

where $H$ is the number of steps. Theoretical results show that the optimal policy $\pi^*$ satisfies the Bellman optimality equation:

$$q^{\pi^*}(s, a) = r(s, a) + \sum_{s' \in \mathcal{S}} T(s'|s, a) \max_{a'} q^{\pi^*}(s', a'). \tag{2}$$

For text generation, the MDP state can be defined as the partial generated sequence $\boldsymbol{y}_{<t} := (y_1, \cdots, y_{t-1})$, and the action as the next token $y_t$ in the vocabulary $\mathcal{V}$. The transition dynamic $T(\cdot|s, a)$ here is deterministic, since every state–action pair $(\boldsymbol{y}_{<t}, y_t)$ leads to a unique state $\boldsymbol{y}_{<t+1}$ for the next step.

In previous RL-based text generation, there lacks a naturally defined reward function $r(s, a)$. While researchers have applied various heuristics as the reward [1, 50], they suffer from several shortcomings (e.g., sparsity and task specificity) as mentioned in Section 1. To address these problems, we propose to induce a reward function for text generation tasks in a principled approach by inverse reinforcement learning.

**Inverse Reinforcement Learning (IRL).** The goal of IRL is to learn a reward function $r(s, a)$. Especially, we wish the resulting action value function $q$ computed by Eqn. (1) could satisfy $q(s, a) \geq q(s, a')$ for every $a' \in \mathcal{A}$ and every $(s, a)$ pair in the training set $\mathcal{D}$. In other words, the decisions in $\mathcal{D}$ are made greedily by $\operatorname{argmax}_a q(s, a)$ given any state $s$. Unfortunately, Ng and Russell [36] show that this is an ill-posed problem since the desirable reward function $r$ is not unique. Therefore, we follow a common assumption [4, 43, 69] to resolve the ambiguity:

**Assumption 1.** Given an action value function $q$, the policy $\pi$ takes the form of $\pi_q(a|s) := \exp(q(s, a)) / \sum_{a'} \exp(q(s, a'))$.

In traditional IRL [4, 43, 69], reward learning is difficult and this assumption does not directly yield a reward function due to the stochastic state transition $T(s'|s, a)$. However, our insight is that the transition is deterministic for text generation tasks, and thus we may utilize Assumption 1 to induce an action value function $q$, and then a reward function $r$, from some learned policy $\pi$, as explained in the next part.

## 2.2 Teacher Forcing Recovers IRL

One of our main contributions is that we show the seemingly complicated reward learning in Section 2.1 can be recovered by teacher forcing, the *de facto* common practice of supervised text generation. Our discovery leads to a convenient approach that derives a step-wise reward function simply from general seq2seq models, without the need for task-specific heuristics. This makes RL more general for text generation, and our step-wise reward largely simplifies RL training.

**Maximum Likelihood Estimation (MLE) for IRL.** Following Assumption 1, we let the policy $\pi_{q_\omega}(\cdot|s) \propto \exp(q_\omega(s, \cdot))$, where $q_\omega$ is a parameterized action value function. Under such a policy, the probability of each trajectory $\tau := ((s_1, a_1), \ldots, (s_{|\tau|}, a_{|\tau|}))$ in the dataset is given by the trajectory distribution $P^{\pi_{q_\omega}}$. The likelihood of the dataset is given by

$$P_{\text{IRL}}(\mathcal{D}|\omega) := \prod_{\tau \in \mathcal{D}} P^{\pi_{q_\omega}}(\tau). \tag{3}$$

**Teacher-Forcing Training.** For text generation, the standard teacher-forcing seq2seq training is to minimize the loss:

$$L_{\text{TF}}(\omega; \mathcal{D}) := - \sum_{\boldsymbol{y} \in \mathcal{D}} \sum_{t=1}^{|\boldsymbol{y}|} \log p_\omega(y_t|\boldsymbol{y}_{<t}), \tag{4}$$

where the predicted probability of the next token being $v$ is $p_\omega(v|\boldsymbol{y}_{<t}) = \frac{\exp(f_\omega(\boldsymbol{y}_{<t}, v))}{\sum_{v' \in \mathcal{V}} \exp(f_\omega(\boldsymbol{y}_{<t}, v'))}$ for the logit function $f_\omega$ with parameters $\omega$. In seq2seq training, an additional input $\boldsymbol{x}$ may be added to the conditional probabilities but is omitted here for simplicity.

The below theorem shows their equivalence up to an additional constant.

**Theorem 1.** *Suppose the value function $q$ in Eqn. (3) and the seq2seq model $f$ in Eqn. (4) have the same parametrization $\omega$, we have*

$$L_{\text{TF}}(\omega; D) = -\log P_{\text{IRL}}(\mathcal{D}|\omega) + \text{const}. \tag{5}$$

*Proof.* For the MLE of IRL under Assumption 1, the Ionescu–Tulcea theorem [22] asserts that there exists a unique trajectory distribution $P^\pi_\mu$ satisfying

$$P^\pi_\mu(s_1) = \mu(s_1),$$
$$P^\pi_\mu(s_1, a_1, \ldots, s_t, a_t) = P^\pi_\mu(s_1, a_1, \ldots, s_t)\pi(a_t|s_t),$$
$$P^\pi_\mu(s_1, a_1, \ldots, s_t, a_t, s_{t+1}) = P^\pi_\mu(s_1, a_1, \ldots, s_t, a_t)T(s_{t+1}|s_t, a_t)$$

for any $t \geq 1$, given the initial state distribution $\mu$, transition probability $T$, and policy $\pi$.

The likelihood can thus be factorized by the multiplication of $\mu$, $T$, and $\pi$:

$$P_{\text{IRL}}(\mathcal{D}|\omega) = \prod_{\tau \in \mathcal{D}} P^{\pi_{q_\omega}}_\mu(\tau) = \prod_{\tau \in \mathcal{D}} \left[ \mu(s_1)\pi_{q_\omega}(a_1|s_1) \prod_{t=2}^{|\tau|} T(s_t|s_{t-1}, a_{t-1})\pi_{q_\omega}(a_t|s_t) \right].$$

As mentioned, text generation has a deterministic transition, i.e., $T(s'|s, a) = 1$ for the next state $s' = s + [a]$. Taking the $\mu$ terms out, we have

$$-\log P_{\text{IRL}}(\mathcal{D}|\omega) = -\log \prod_{\tau \in \mathcal{D}} \prod_{t=1}^{|\tau|} \pi_{q_\omega}(a_t|s_t) - \log \prod_{\tau \in \mathcal{D}} \mu(s_1), \tag{6}$$

where the second term is a constant in terms of $\omega$. In Section 2.1, text generation is modeled as an MDP with $s_t = \boldsymbol{y}_{<t}$ and $a_t = y_t$. Therefore, the first term of Eqn. (6) is the same as Eqn. (4) under the parametrization $\pi_{q_\omega} = p_\omega$, concluding the equivalence between MLE for IRL and the teacher-forcing training of a seq2seq model. □

**Inducing the Reward Function.** Theorem 1 shows that seq2seq training with teacher forcing actually learns an IRL model. Thus, we may derive a reward function assuming the action value function is well trained:

$$r(s,a) = q_\omega(s,a) - \sum_{s' \in \mathcal{S}} T(s'|s,a) \max_{a' \in \mathcal{A}} q_\omega(s',a') = f_\omega(s,a) - \max_{a' \in \mathcal{A}} f_\omega(s+[a],a'), \quad (7)$$

where the first equality is due to the Bellman optimality condition (2); the second equality is due to the parametrization of $q_\omega = f_\omega$ and the deterministic transition $T(s'|s,a) = 1$ for $s'$ being the concatenation of the prefix $s$ and token $a$.

*Remark* 1. It is easy to notice that $f_\omega(s,\cdot)$ may be arbitrarily shifted by a constant $c_s$ without changing $\pi_\omega$. This also shifts the derived reward $r(s,a)$ by $c_s - c_{s+[a]}$. However, it does not affect the optimal policy. We will prove this in Theorem 3 after introducing policy gradient methods.

Our use of the Bellman optimality condition is different from classic RL, where the reward is well-defined and the action value function is thus learned [55]. Instead, we induce the underlying reward assuming the action value function is known (given by Assumption 1). The following diagram shows the whole process of our derivation.

$$\mathcal{D} \xrightarrow{\text{Teacher Forcing}} \pi \xrightarrow{\text{Assumption 1}} q \xrightarrow{\text{Eqn. (7)}} r$$

In real-world applications, the learned action value function might be imperfect; in this case, we may bound the error of our induced reward with the following theorem.

**Theorem 2.** *Let $r^*$ be an underlying true reward function and $q^*$ be the corresponding optimal value function. Given an approximate value function $q$, we denote by $r$ the reward function derived from Eqn. (7). Then, we must have $\|r - r^*\|_\infty$ bounded by $O(\|q - q^*\|_\infty)$. Here, $\|\cdot\|_\infty$ takes the maximum absolute value over all $s \in \mathcal{S}$ and $a \in \mathcal{A}$.*

*Proof.* See Appendix A. ☐

## 2.3 Periodically Synchronized Behavior Policy in Policy Gradient

In text generation, a neural network can be viewed as a policy $\pi$ that predicts the word distribution given the state of a decoding step. The reward induced from Section 2.2 can be used to improve the policy through RL. To stabilize training, we propose a variant of off-policy policy gradient methods [10] with a periodically synchronized behavior policy.

Our RL training adopts the off-policy REINFORCE [63] as the backbone of our algorithm. Let $\pi_\varphi$ be the model policy (i.e., the model's prediction) to be optimized, and $\pi_b$ be the behavior policy (i.e., the sampling distribution during training). Through importance sampling, the gradient of the expected total reward with respect to $\varphi$ can be obtained by the off-policy policy gradient theorem [10]

$$\nabla_\varphi \mathop{\mathbb{E}}_{\pi_\varphi} \left[ \sum_t r(s_t, a_t) \right] = \mathop{\mathbb{E}}_{\pi_b} \left[ \sum_t \rho_t \hat{q}_r(s_t, a_t) \nabla_\varphi \log \pi_\varphi(a_t|s_t) \right], \quad (8)$$

where $\rho_t := \pi_\varphi(a_t|s_t)/\pi_b(a_t|s_t)$ is the importance weight, and $\hat{q}_r(s_t, a_t) := \sum_{i \geq t} r(s_i, a_i)$ is the total reward of the trajectory. In practice, off-policy REINFORCE ($\pi_\varphi \neq \pi_b$) is more exploratory than the on-policy one ($\pi_\varphi = \pi_b$), since the model policy $\pi_\varphi$ would become more concentrated during optimization and does not explore much, whereas $\pi_b$ is typically chosen to cover more trajectories. However, Degris et al. [10] adopt a fixed behavior policy $\pi_b$, which does not perform exploitation according to the current model policy. The lack of exploitation might lead to less informative training.

To balance exploration and exploitation, we would like the behavior policy to be close to the model policy but stay exploratory at the same time. We thus propose a periodically updating schedule, where the behavior policy is frozen for a long period to encourage exploration but keeps track of the current model policy to enhance exploitation. Particularly, we synchronize the behavior policy with the model policy for every $k$ gradient updates of the latter (e.g., $k = 5000$). Our remedy is a simple method overcoming the instability of REINFORCE. It shares a common ground with a number of policy gradient methods like the proximal policy optimization (PPO) [48], especially in that both

methods involve multiple updates with a fixed behavior policy. As the main contribution of this paper is reward induction, we resort to this simple fix and leave the mathematical connection as an interesting future direction.

Algorithm 1 summarizes our approach. Our implementation is able to execute the loops in parallel, which speeds up the training process. Our periodically synchronized behavior policy further enables us to parallelize sampling and model updates to reduce the awaiting time.

## 2.4 Application to Semi-Supervised Learning

Our approach naturally aligns with the paradigm of semi-supervised learning, as it involves training a seq2seq model to induce the reward function, which requires (at least a small volume of) parallel data $\mathcal{D}_p$. Additionally, we assume there is a non-parallel dataset $\mathcal{D}_u$ containing input sentences only for RL training with the induced reward.

Our semi-supervised approach consists of two stages. We first train a seq2seq model $f_\omega$ on the parallel dataset $\mathcal{D}_p$ to induce the reward function $r$ by Eqn. (7). The procedure is described in Section 2.2. The reward function then facilitates RL training on the non-parallel dataset $\mathcal{D}_u$, which is shown in Algorithm 1.

---

**Algorithm 1:** Our Algorithm

**Input:** A non-parallel dataset $\mathcal{D}_u$, learned logit (value) function $f_\omega$, policy $\pi_\varphi$ with the initial parameter $\varphi$, total update steps $U$, and synchronizing period $k$

**Output:** A policy $\pi_\varphi$ parameterized by $\varphi$

**begin**
 **for** $i \leftarrow 1...U$ **do**
  **if** $i \equiv 0 \pmod{k}$ **then**
   $\pi_b \leftarrow \pi_\varphi$ ;         ▷ Behavior policy update
  Sample a source sentence $\boldsymbol{x} \in \mathcal{D}_u$
  Construct the initial state $s \leftarrow (\boldsymbol{x}, [\text{BOS}])$ ; ▷ [BOS] is the beginning token
  Sample a trajectory $\tau$ from the behavior policy $\pi_b$
  $\hat{q}_{h+1}^r \leftarrow 0$ and $g \leftarrow \mathbf{0}$
  **for** $t \leftarrow |\tau|...1$ **do**
   **if** $t = |\tau|$ **then**
    $r_t \leftarrow f_\omega(s_t, a_t)$ ;      ▷ Termination step, no $s_{t+1}$
   **else**
    $r_t \leftarrow f_\omega(s_t, a_t) - \max_{a'} f_\omega(s_{t+1}, a')$ ;   ▷ By Eqn. (7)
   $\hat{q}_t^r \leftarrow r_t + \hat{q}_{t+1}^r$ ;     ▷ Accumulating rewards
   $\rho_t \leftarrow \pi_\varphi(a_t|s_t)/\pi_b(a_t|s_t)$ ;    ▷ Importance weight
   $g \leftarrow g + \rho_t \hat{q}_t^r \nabla_\varphi \log \pi_\varphi(a_t|s_t)$ ;    ▷ By Eqn. (8)
  $\varphi \leftarrow \varphi + \eta g$ ;       ▷ Gradient ascent
 **return** $\pi_\varphi$

---

As mentioned in Remark 1, the reward $r(s, a)$ can be arbitrarily shifted by $c_s - c_{s+[a]}$. We show that this shift does not affect the optimal policy.

**Theorem 3.** *Suppose* $r'(s, a) = r(s, a) + c_s - c_{s+[a]}$. *Then the learned policies under* $r'(s, a)$ *and* $r(s, a)$ *are the same.*

*Proof.* By Eqn. (1), function $q$ returns the expected total reward. In Algorithm 1, we sample it by

$$\hat{q}_t^{r'}(s_t, a_t) := r'(s_t, a_t) + r'(s_{t+1}, a_{t+1}) + \cdots + r'(s_{|\tau|}, a_{|\tau|})$$
$$= r(s_t, a_t) + c_{s_t} - \cancel{c_{s_{t+1}}} + r(s_{t+1}, a_{t+1}) + \cancel{c_{s_{t+1}}} - c_{s_{t+2}} + \cdots + r(s_{|\tau|}, a_{|\tau|}) + \cancel{c_{s_{|\tau|}}}$$
$$= c_{s_t} + r(s_t, a_t) + r(s_{t+1}, a_{t+1}) + \cdots + r(s_{|\tau|}, a_{|\tau|}) =: \hat{q}^r(s_t, a_t) + c_{s_t}.$$

The last line suggests the constant plays a role as the baseline in policy gradient, which is shown to be irrelevant to the optimal policy [55]. □

Table 1: Main results. $^{\uparrow/\downarrow}$The higher/lower, the better. $^{\dagger}$Quoted from Wen et al. [62] on deduplicated dialogue datasets. $^{\ddagger}$Quoted from [29]. $^{\S}$Quoted from [11]. For the paraphrase generation metric, we have iBLEU $= (1 - \alpha)$ BLEU $-\alpha$ SBLEU.

(a) Dialogue generation.

| Method | BLEU2$^{\uparrow}$ | BLEU4$^{\uparrow}$ |
|---|---|---|
| Parallel DailyDialog | | |
| AdaLabel$^{\dagger}$ [60] | 6.72 | 2.29 |
| DialogBERT$^{\dagger}$ [16] | 5.42 | 2.16 |
| T5-Base [42] | **8.96** | **3.69** |
| + Parallel OpenSubtitles | | |
| [T5-Base] Fully Supervised | 8.75 | 3.06 |
| + Non-Parallel OpenSubtitles | | |
| [T5-Base] Self-Training | 9.10 | 3.73 |
| [T5-Base] R-Regression | 10.34 | 4.18 |
| [T5-Base] Ours | **11.02** | **4.30** |

(b) Paraphrase generation. "Copy" refers to direclty copying the input sentence.

| Method | BLEU4$^{\uparrow}$ | SBLEU4$^{\downarrow}$ | iBLEU4$^{\uparrow}$ |
|---|---|---|---|
| Copy | 29.88 | 100.0 | 16.89 |
| Parallel Quora Generation | | | |
| Dagger$^{\ddagger}$ [12] | 28.42 | 66.98 | 18.88 |
| RL-NN$^{\ddagger}$ [40] | 20.98 | **40.52** | 14.83 |
| T5-Base [42] | **30.83** | 44.77 | **23.27** |
| + Non-Parallel Quora Generatoin | | | |
| LTSL$^{\S}$ [11] | 29.25 | 71.25 | 19.20 |
| [T5-Base] Self-Training | 31.39 | 48.02 | 23.44 |
| [T5-Base] R-Regression | 30.77 | **44.23** | 23.27 |
| [T5-Base] Ours | **31.47** | 45.43 | **23.78** |

## 3 Experiments

### 3.1 Datasets and Metrics

**Dialogue Generation.** We adopt two widely used datasets, DailyDialog [30] and OpenSubtitles [57], for the dialogue experiment. The DailyDialog dataset is constructed from English dialogues crawled from the Internet, whereas the OpenSubtitles dataset is constructed from movie subtitles based on IMDB identifiers. A dialogue session is split into single-turn context–response pairs in our experiment. For semi-supervised learning, we use the smaller dataset, DailyDialog, as the parallel corpus $\mathcal{D}_p$, and the larger dataset, OpenSubtitles, as the non-parallel corpus $\mathcal{D}_u$ (i.e., we only retain the context sentence in the OpenSubtitles dataset). This follows the common setup for semi-supervised learning, where the unlabeled dataset is larger than the labeled one.

It should be emphasized that a recent study [62] shows more than 20% of test samples are identical to some training samples in both DailyDialog and OpenSubtitles. This results in meaningless comparison and inflated performance of previous methods, e.g., a BLEU4 of 11.01 in AdaLabel [60] and 14.61 in DialogBERT [16]. Therefore, we use the deduplicated datasets in [62], containing 60K/6.5K/7K samples for training/validation/test in DailyDialog and 1M non-parallel samples in OpenSubtitles. Although our scores will be lower than previous inflated ones, we follow the correct setting for research.

We use BLEU scores [38] as main evaluation metrics, which are widely used in dialogue generation [62]. In particular, BLEU-$n$ evaluates the geometric average of $i$-gram precision scores for $i = 1, \cdots, n$. Following previous work [62], we lowercase all sentences and tokenize them with the NLTK library [33].

**Paraphrase Generation.** We follow previous studies [11, 32, 34] and use the Quora Question Pair dataset[2] for the paraphrasing experiment. The Quora dataset is originally designed for paraphrase classification, containing both paraphrase and non-paraphrase pairs. The paraphrase pairs naturally form a parallel dataset for the generation purpose; following the common practice [34], we split it into 124K/4K/20K samples for training/validation/test. The non-paraphrase pairs, containing 510K sentences, are discarded in previous work, but we are able to utilize them in a semi-supervised manner.

We use the standard iBLEU score [54] as the main evaluation metric. It involves a penalty of Self-BLEU (SBLEU) between the generated and input sentences, as the paraphrasing task requires using

---

[2] https://www.kaggle.com/c/quora-question-pairs

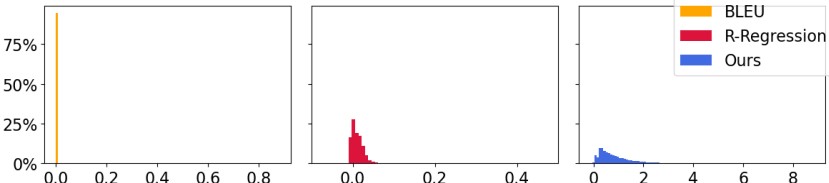

Figure 1: The distributions of token-level estimation of future rewards ($\hat{q}_t^r$ in Algorithm 1) on the DailyDialog validation set. The BLEU score of a sentence is shared among tokens in the same sentence.

different lexicons. Specifically, it is calculated by iBLEU = $(1 - \alpha)$ BLEU $-\alpha$ SBLEU, where $\alpha$ is typically set to 0.1 [11, 32, 34]. For clarity, we also report BLEU and S-BLEU scores in our experiment.

## 3.2 Settings and Competing Methods

For each task, we first fine-tune a T5-Base model [42] on the parallel data by Eqn. (4). Then we apply our proposed method to induce the reward and further train the model by Algorithm 1 on the non-parallel data. We compare our approach with the following semi-supervised methods.

**Self-Training.** We apply the supervised model to the non-parallel dataset and generate pseudo-target sentences, which are used to continue training the model. This is a commonly used semi-supervised approach in text generation literature [21, 68].

**R-Regression.** Wu et al. [66] propose a reward regression (R-Regression) approach, where the reward is defined as the BLEU score. Since their reward is the same as the evaluation metric, such a method may achieve higher BLEU scores without actually improving the generation quality. By contrast, our reward is induced in a principled way and is agnostic to evaluation metrics. In our experiment, we replicate the R-regression method, which constitutes a controlled comparison to our approach, as the only difference is the reward function.

Appendix B provides implementation details and hyperparameters of our approach.

## 3.3 Main Results

**Results of Dialogue Generation.** Table 1a shows the results of the dialogue generation task. We notice that our fine-tuned T5-Base model [42] has already outperformed dedicated methods, AdaLabel [60] and DialogBERT [16]. This is consistent with the findings of [62, 61] in that the alleged "state-of-the-art" dialogue systems do not outperform standard pretrained language models on deduplicated datasets, highlighting the importance of working with the correct setting.

We then apply semi-supervised learning (Self-Training, R-Regression, and our approach) with the non-parallel OpenSubtitles dataset. We achieve higher performance than T5-Base trained only on parallel DailyDialog. Interestingly, the fully supervised model—trained on both parallel DailyDialog and parallel OpenSubtitles—does not achieve high performance, even lower than the one trained with DailyDialog only. It is noticed that the OpenSubtitles dataset is noisy [8], which likely causes the performance degradation. This signifies the need of semi-supervised learning.

Among semi-supervised approaches, RL-based methods (R-Regression and ours) are generally better than Self-Training. This is within our expectation because Self-Training learns from its own generation and may be overconfident, whereas RL approaches are able to explore different parts of the data space, being a more effective way of semi-supervised learning.

Moreover, our approach outperforms RL with R-Regression, where the reward is the only difference. The controlled experiment confirms that the reward induced from models trained with teacher forcing is effective for RL training. It is also worth noting that R-Regression uses the evaluation metric as the reward, and thus may deliberately improve the metric rather than text quality. By contrast, our reward is induced in a principled manner and is agnostic to evaluation metrics, and our approach still achieves higher performance even with such a disadvantage.

Table 2: Comparing sparse and dense reward functions.

(a) Dialogue generation.

| Sparse | Method | BLEU2$^\uparrow$ | BLEU4$^\uparrow$ |
|---|---|---|---|
| - | Self-Training [23] | 9.10 | 3.73 |
| Yes | R-Regression [66] | 9.45 | 3.73 |
| | Induced-R | **9.75** | **3.99** |
| No | R-Regression [66] | 10.34 | 4.18 |
| | Induced-R | **11.02** | **4.30** |

(b) Paraphrase generation.

| Sparse | Method | BLEU4$^\uparrow$ | SBLEU$^\downarrow$ | iBLEU4$^\uparrow$ |
|---|---|---|---|---|
| - | Self-Training [23] | 31.39 | 48.11 | 23.44 |
| Yes | R-Regression [66] | 30.78 | **44.32** | 23.27 |
| | Induced-R | **31.28** | 45.22 | **23.63** |
| No | R-Regression [66] | 30.77 | **44.23** | 23.27 |
| | Induced-R | **31.47** | 45.43 | **23.78** |

In general, our approach achieves the best performance in both metrics. In particular, it significantly improves DailyDialog-trained T5-Base by +2.06 (+23.0%) in BLEU2 and +0.61 (+16.5%) in BLEU4. It also outperforms the second-best method, R-Regression, by 0.68 (+6.6%) in BLEU2 and 0.12 (+2.9%) in BLEU4, verifying the effectiveness of our approach.

**Results of Paraphrase Generation.** The results of paraphrase generation are shown in Table 1b. As seen, directly copying the input already achieves a high BLEU score against the reference. iBLEU addresses this by penalizing the Self-BLEU score (against input) and is considered the main metric.

We consider another semi-supervised baseline LTSL [11]. It performs retrieval-based paraphrase expansion and meta optimization, thus being task specific. We see that LTSL has an extremely high Self-BLEU, suggesting the generated paraphrase largely resembles the input. It achieves a lower iBLEU score than other semi-supervised approaches.

We also see that RL approaches generally achieve lower Self-BLEU than Self-Training. This is because Self-Training learns from its own predictions, which overlap the input more than groundtruth paraphrases do (Self-BLEU of groundtruth: 29.87); as a result, Self-BLEU increases to 48.02 from 44.77 of T5-Base. By contrast, RL learns by exploring different possible paraphrases and is able to retain low Self-BLEU.

Overall, our approach achieves the highest BLEU and a reasonably low Self-BLEU, yielding the best iBLEU among all competing methods. The results are consistent with Table 1a, showing the generality of our approach.

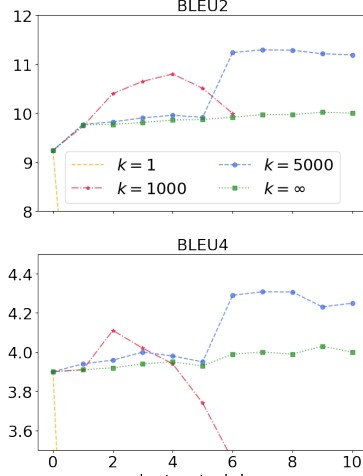

Figure 2: The learning curves by choosing different values of $k$. Scores are measured on the validation set of DailyDialog. Training is terminated when the BLEU4 score drops below 3.5.

### 3.4 Analyses

**Step-Wise Reward.** In Figure 1, we show the distributions of different reward functions. As seen, the BLEU score is mostly concentrated at 0, providing little information for training. R-Regression consequently suffers from a similar problem, as it is trained by the groundtruth BLEU scores. The distribution of our induced reward, on the other hand, has the lowest peak and is the most wide-spreading one.

We conduct another analysis to show the importance of step-wise rewards for RL training. We compare our approach with a sparse reward function that defers all rewards to the end of a sentence. In other words, the last step's reward is the sum of our step-wise rewards, whereas all previous steps have a reward of 0. This constitutes a rigorous analysis, as the total reward and thus the training objective are the same in both cases. Results in Table 2 show that our step-wise reward outperforms the sparse reward in all cases. This suggests our approach serves as a meaningful credit assignment of the total reward, which is beneficial for RL training.

**The Effect of the Synchronizing Period.** We analyze the effect of the synchronizing period $k$ introduced in Section 2.3. In Figure 2, we see that the training is unstable if $k = 1$ (on-policy), in which case the model generates uninformative and meaningless sentences (illustrated in Appendix D). When $k = 1000$, the performance increases quickly at the beginning, but it starts to decrease with further training. We hypothesize that this is due to the lack of exploration (Section 2.3). When $k$ is infinitely large (the behavior policy is fixed), the performance grows slowly and stops improving after a certain number of steps. Based on this analysis, we choose $k = 5000$ to balance exploitation and exploration. Although the experiment is conducted only on DailyDialog due to the limit of time and resources, we directly apply the setting to other experiments, showing the robustness of our approach.

**Data Efficiency.** In Figure 3, we analyze data efficiency by sampling different numbers of data points from the non-parallel corpus. As shown, our method consistently outperforms self-training, even with only 0.1% (the leftmost points) of the training set. Additionally, the performance of our method quickly increases with more data, whereas self-training grows slowly. This is expected because RL training explores different parts of the sentence space and learns from their rewards, whereas self-training only learns from the single generated sentence by the model itself given an input.

We also investigate how performance changes according to the size of the parallel dataset, which reflects the quality of the learned policy. Results are shown in Figure 4, Appendix C.

## 4 Related Work

**Semi-Supervised Learning for Text Generation.** In text generation, popular ways to utilize both parallel and non-parallel data include self-training [21, 68] and back-translation [49]. Both methods first train a model on the parallel data and then generate pseudo-parallel pairs for the non-parallel sentences. The difference is that self-training generates pseudo-parallel pairs from source to target, whereas back-translation generates from target to source. We mainly consider self-training as a baseline because it does not require an additional model in the reversed direction, making the comparisons fairer. Our implementation of self-training is also similar to sequence-level knowledge distillation [23, 15, 20], except that the latter augments the parallel data instead of the non-parallel ones. In Figure 3, we show that self-training cannot efficiently utilize the data because of the lack of exploration. Additionally, the exposure bias issue remains because they are trained with the teacher-forcing objective.

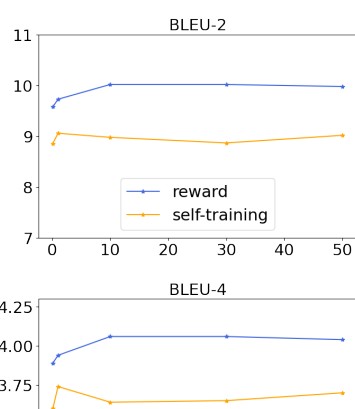

Figure 3: Trends of self-training and our method given different sizes of the non-parallel data. Scores are measured on the Daily-Dialog test set.

**Text Generation beyond Teacher Forcing.** Teacher forcing is known to have the exposure bias issue. A line of work uses the generative adversarial network (GAN) [14] to alleviate the issue. For example, Yu et al. [67] and Guo et al. [18] propose to use GAN-style training to generate text similar to the training set in an on-the-fly manner. This practice reduces the discrepancy between training and inference because GAN sends its own generation as inputs rather than using groundtruth sentences during training. Shi et al. [51] further formulate the adversarial training using the IRL interpretation. These GAN-style methods are different from ours in two main ways. First, GAN-style training requires parallel corpora and thus cannot be directly applied to semi-supervised learning on non-parallel datasets. Second, GAN-style training involves the optimization of an adversarial objective, making the training unstable, e.g., suffering from mode collapse [14].

Another paradigm to alleviate the exposure bias is RL. For instance, Sokolov et al. [53] and Kreutzer et al. [26] leverage the bandit-structured prediction framework for text generation with BLEU as the heuristically defined reward. Bahdanau et al. [1] and Shen et al. [50] utilize different variants of policy gradient for RL training. However, these methods are task-specific and suffer from the problem of sparse rewards, as mentioned in Section 3.4. More importantly, these approaches require

parallel data to calculate the reward and cannot utilize non-parallel data either. To address this, Wu et al. [66] propose to learn a reward regression model on the parallel dataset and perform RL on the non-parallel data with the learned reward. As mentioned, such a method is still task-specific because it requires the human heuristics of the task to define the proper reward function. Additionally, it suffers from the reward-sparsity problem, as seen in Figure 1 and Table 2.

Search is also a popular way to replace teacher forcing. The Learning to Search (L2S) framework [5, 9] enables the model to search for a better score during learning and is widely applied to text generation. For example, Wiseman and Rush [64] propose to optimize the beam search results through training. Li et al. [29] develop an unsupervised learning approach to text generation based on local search. In addition to the L2S framework, Leblond et al. [27] leverage the Monte Carlo tree search [25, 52] to select better tokens in a step from the sampled generation. These methods are different from ours since they need either heuristically defined scoring functions or parallel data, limiting their methods to certain tasks or to the supervised paradigm. However, given the success of these methods, we consider the search-based approach an interesting future extension of our work.

**Imitation Learning.**    The intuition behind our work is also related to imitation learning methods in general. Typically, these methods aim to obtain a good policy given a dataset containing state–action pairs. The easiest approach is behavior cloning [39], which greedily imitates the demonstration. Similar to the exposure bias, behavior cloning also faces the problem of compounding errors [46]. SMILe [46] and DAgger [47] mitigate the problem by querying an expert. In text generation, Du and Ji [12] empirically verify that imitation learning methods are helpful. Recently, Pang and He [37] frame the text generation task as an offline reinforcement learning problem, which learns from a dataset containing tuples of state, action, and reward. Compared with our method, these approaches rely on parallel sentence pairs and cannot effectively make use of non-parallel datasets.

## 5   Conclusion

**Summary.**    In this paper, we show that a reward function can be derived from a model trained with teacher forcing. The derivation does not rely on human heuristics for certain tasks. Additionally, the derived reward function assigns step-wise scores and makes the RL training easier. Our approach leads to a training algorithm in a semi-supervised manner and utilizes both parallel and non-parallel data. We conduct experiments on the dialogue and paraphrase generation tasks. The empirical results show that the performance of our approach is better compared with the baselines: self-training and reward regression. We further analyze our reward function and show the benefits of our approach.

**Limitation and Future Work.**    First, the scale of the experiments in this paper is restricted by computational resources. It is interesting to see if our approach could obtain better performance with large models [3, 41] and larger datasets.

We also notice that Assumption 1 has a deep connection with entropy-regularized RL [17, 19, 45]. Our approach can be easily extended to such cases in the future.

Another interesting direction would be using the reward as an interface between humans and the model to control the generation. Specifically, the current seq2seq models treat data as the ground truth, but the data may be contaminated with undesired or harmful information. We hope that our approach provides a way for humans to apply additional rules to the reward function to avoid the model generating harmful information.

## Appendices

The full paper, including appendices, is available at `https://arxiv.org/abs/2210.08708`.

## Acknowledgments

We thank all reviewers for their valuable comments. We also thank Guoqing Luo for early discussions. The research is supported in part by the Natural Sciences and Engineering Research Council of Canada (NSERC) under grant No. RGPIN2020-04465, the Amii Fellow Program, the Canada CIFAR AI Chair Program, a UAHJIC project, a donation from DeepMind, and the Digital Research Alliance of Canada (alliancecan.ca).

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
