# A   Proof of Theorem 2

**Theorem 2.** *Let $r^*$ be an underlying true reward function and $q^*$ be the corresponding optimal value function. Given an approximate value function $q$, we denote by $r$ the reward function derived from Eqn. (7). Then, we must have $\|r - r^*\|_\infty$ bounded by $O(\|q - q^*\|_\infty)$. Here, $\|\cdot\|_\infty$ takes the maximum absolute value over all $s \in \mathcal{S}$ and $a \in \mathcal{A}$.*

*Proof.* For any $s \in \mathcal{S}$ and $a \in \mathcal{A}$, we have

$$|r(s,a) - r^*(s,a)|$$
$$=|q(s,a) - q^*(s,a) + \max_{a'} q^*(s+[a],a') - \max_{a'} q(s+[a],a')| \tag{9}$$

$$\leq |q(s,a) - q^*(s,a)| + |\max_{a'} q^*(s+[a],a') - \max_{a'} q(s+[a],a')| \tag{10}$$

$$\leq \max_{s',a'} |q(s',a') - q^*(s',a')| + \max_{s'} |\max_{a'} q^*(s',a') - \max_{a'} q(s',a')| \tag{11}$$

$$\leq \max_{s',a'} |q(s',a') - q^*(s',a')|$$
$$+ \max_{s'} \max\{\max_{a'} q^*(s',a') - \max_{a'} q(s',a')), \max_{a'} q(s',a') - \max_{a'} q^*(s',a))\} \tag{12}$$

$$\leq \max_{s',a'} |q(s',a') - q^*(s',a')|$$
$$+ \max_{s'} \max\{\max_{a'}(q^*(s',a') - q(s',a')), \max_{a'}(q(s',a') - q^*(s',a'))\} \tag{13}$$

$$\leq 2 \max_{s',a'} |q(s',a') - q^*(s',a')| \tag{14}$$

$$=2\|q - q^*\|_\infty. \tag{15}$$

Here, Eqn. (9) is from the Bellman equation; Eqn. (10) follows the triangle inequality; and Eqn. (11) generalizes certain $s$ and $a$ to all possible $s' \in \mathcal{S}, a' \in \mathcal{A}$. Eqn. (12) discusses two possible cases: whether $\max_{a'} q(s',a') \geq \max_{a'} q^*(s',a')$ or not. Eqn. (13) is because $-\max_{a'} q(s',a') \leq -q(s',a'')$ for any $a'' \in \mathcal{A}$. Eqn. (14) merges all the maximum operation, and Eqn. (15) is the definition of the infinity norm.

Since the last equation does not depend on $s$ and $a$, we conclude $\|r - r^*\|_\infty$ is bounded by $O(\|q - q^*\|_\infty)$. □

# B   Experiments Details

For all experiments, we initialize the model with T5-Base [42] provided by HuggingFace [65]. We use the label smoothing [56] with a coefficient of $0.1$. We use the Adam [24] optimizer with $(\beta_1, \beta_2) = (0.9, 0.999)$. Each batch contains around 32K tokens.

For all conventional seq2seq training, the learning rate is scheduled according to the original Transformer [58] with the warm-up steps set as $4000$. For all RL training, we drop the warm-up phase and set the maximum learning rate to $1e-5$. We set the synchronizing period $k$ to $5000$. The reward of our method is scaled down by 100 times. We apply the reward clipping trick [35] to bound the reward within $[-1, 1]$ to stabilize the training.

For inference, we follow previous work and use greedy decoding in the dialogue generation task and use beam search with a beam size of $5$ in the paraphrase generation task.

All the experiments are done on either $4\times$NVIDIA A100 or $4\times$NVIDIA V100.

# C   Additional Results

We analyze the effect of the sizes of parallel data in Figure 4. Our approach consistently outperforms competing methods in all settings. The results show that a high-quality $f_\omega$ indeed leads to better performance, but our model is still robust when $f_\omega$ is trained with limited data. Notably, our method drops by 6.8% when having 10% of the parallel data, whereas R-Regression drops by 10.6%. This show that our reward induction approach utilizes the parallel data more effectively.

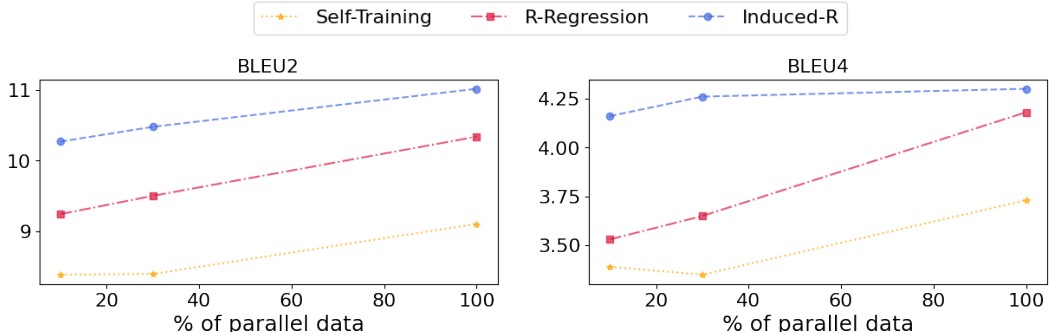

Figure 4: Results of different methods given different sizes of the parallel data. Scores are measured on the DailyDialog test set.

# D    Case Study

We demonstrate several cases from the generation of different models. These cases come from the DailyDialog validation set.

**Examples of Generated Dialogue Responses.**    In the first case of Table 3, we show a phenomenon that previous methods tend to generate short and meaningless responses. On the other hand, our method usually generates more informative sentences and makes the conversation more natural and human-like.

Table 3: Examples of generated dialogue responses.

| | | |
|---|---|---|
| Context | | We can make shipment within one month from receipt of order. |
| Response | Self-Training | I see. |
| | R-Regression | I see. |
| | Ours | I see. I'll have to discuss it with my manager. |
| Context | | Where's your girlfriend? I thought you were going out with her today. |
| Response | Self-Training | I got engaged. We broke up last week. |
| | R-Regression | I got engaged. She told me she's just married. |
| | Ours | She came back from Australia last week. She is a nice girl but there's nothing I can do about her. |

We also find that previous methods tend to generate sentences with inconsistent or even conflicting semantics. In the second case in Table 3, for example, both Self-Training and R-Regression reply "I got engaged" but the next sentences are illogical. This implies that previous methods may generate low-quality sentences even if they have seemingly decent BLEU scores. By contrast, our model generates a more proper response.

**On-Policy Degeneration.**    In Section 2.3, we mention that if $k = 1$ (on-policy), the generation will become deterministic and uninformative. We show such cases in Table 4. The responses are generated by the first save (1000 updates) of the model in the experiment.

Table 4: Failure cases of on-policy training ($k = 1$).

| | |
|---|---|
| Context | We can make shipment within one month from receipt of order. |
| Response | I see. I'll have to think about it. |
| Context | Where's your girlfriend? I thought you were going out with her today. |
| Response | I'm sorry, but I'm not sure I'll be able to make it. I'll have to think about it. |

For both cases, the model replies "I'll have to think about it" at the end of the sentences. In fact, most of the generated responses end with this phrase, which is redundant and meaningless. This

phenomenon is likely to be a result of over-deterministic and insufficient exploration of the on-policy update. If the behavior policy becomes more deterministic of a certain phrase, it will have a smaller chance to explore other hypotheses. Hence, it will enhance the preferred responses and become even more deterministic. On the contrary, our periodically synchronized behavior policy keeps to be exploratory and does not have the degeneration problem as shown in Table 1 and Figure 2.