# OpenReview forum: "Teacher Forcing Recovers Reward Functions for Text Generation"
_NeurIPS.cc/2022/Conference — NeurIPS 2022 Accept_

### Official Review · Reviewer_bSdL · 2022-06-20

**Rating:** 6
**Confidence:** 3
**Soundness:** 3 good
**Presentation:** 3 good
**Contribution:** 3 good

**Summary:**

This paper investigated the connection between supervised teacher forcing and inverse reinforcement learning. The authors showed that under the same parametrization and a deterministic transition function, the loss function of teacher forcing and maximum entropy IRL are equivalent. Then the authors proposed to use the teacher forcing model to replace the q function in the bellman optimality equation to derive a reward function. Empirical results show that the proposed methods outperform semi-supervised learning baselines on text generation tasks.


-------
POST REBUTTAL:

The authors addressed my concerns. After reading the other reviews and the discussion, I increased the score to 6.

**Questions:**

In terms of experiments, since this paper is about IRL, why not include IRL baselines such as GAN-based IRL algorithms?


**Strengths And Weaknesses:**

Strengths:

The proposed algorithm is interesting. It avoids estimating the partition function (e.g. using GAN or sampling methods) in maximum entropy IRL, but instead using the teacher forcing model to derive a reward function. The training in this case will be more stable.

Weaknesses:

I am not entirely convinced that the proposed reward function in equation (7) is valid.
(1) Theorem 1 in the paper shows that, given the same parameterization, MaxEnt IRL and teacher forcing both maximize the likelihood of the trajectories in the data and hence are equivalent. However, the q function in MaxEnt IRL (in equation (3)) has to additionally satisfy bellman equation and cannot be optimized freely. Therefore, the q function q_w cannot be simply replaced by the logit function f_w, and f_w(s, a) does not equal to q_w(s, a) for a given (s, a) pair. Therefore, I am not entirely convinced that equation (7), which simply substitute q_w with f_w, is valid.

(2) Even if we can replace q_w with f_w, f_w is learned from data and represents the behavior policy, while q_w in equation (7) should be the q function of the optimal policy.

---

> ### Author Response · Authors · 2022-08-01
> **Response to Reviewer bSdL**
>
> We thank the reviewer for saying “the proposed algorithm is interesting” and for highlighting our contributions to IRL in the context of text generation.
>
> ---
>
> > Weakness (1). “Theorem 1 in the paper shows that, given the same parameterization, MaxEnt IRL and teacher forcing both maximize the likelihood of the trajectories in the data and hence are equivalent. However, the q function in MaxEnt IRL (in equation (3)) has to additionally satisfy bellman equation and cannot be optimized freely.” Therefore, the q function q_w cannot be simply replaced by the logit function f_w, and f_w(s, a) does not equal to q_w(s, a) for a given (s, a) pair. Therefore, I am not entirely convinced that equation (7), which simply substitute q_w with f_w, is valid.
>
> Thanks, but this is a misunderstanding. The Bellman equation indeed draws the connection between the reward and the q-value function.
>
> In the RL setting, the reward is given, so we can derive the optimal q-value function by the Bellman optimality equation.
>
> In the IRL setting, the reward is not given. Therefore, it is possible to freely optimize the q-value function, and then use the Bellman optimality equation to derive the underlying reward. Our contribution lies in deriving the q-value function from the policy (based on a common assumption [1, 2, 3] stated in Assumption 1) and then deriving the reward function for text generation, following the IRL setting.
>
> > Weakness (2).  Even if we can replace q_w with f_w, f_w is learned from data and represents the behavior policy, while q_w in equation (7) should be the q function of the optimal policy.“
>
> Thanks for the question. Here, we assume $f_w$ is reasonably trained (as given by teacher-forcing training). Thus, it can be thought of as an approximation to the optimal $q_w$. We will clarify in the revision.
>
> In fact, we additionally derived the theoretical relationship between a near-optimal $f_w$ and a near-optimal $r$. We provided the theorem and the proof in the updated manuscript (last page of  [the submission](https://openreview.net/pdf?id=1_gypPuWUC3)).
>
> This further confirms the validity of our approach.
>
> > Question: In terms of experiments, since this paper is about IRL, why not include IRL baselines such as GAN-based IRL algorithms?
>
> GAN-based IRL requires groundtruth target sequences, so that a discriminator can distinguish between groundtruth and generated ones.
>
> However, our focus is semi-supervised learning for conditional text generation, where we only have a small set of parallel data, but there exists a large amount of non-parallel text. GAN-based IRL is inapplicable to such a semi-supervised setting. We discussed this in Lines 287-291 and will clarify more.
>
> ---
>
> In general, the reviewer considers our contribution “interesting”, but has certain concerns about IRL. We find that these concerns are due to misunderstandings. We hope our response could clarify the confusion. Thank you.
>
> [1] Ramachandra et al. Bayesian inverse reinforcement learning. In IJCAI, 2007.
>
> [2] Ziebart et al.. Maximum entropy inverse reinforcement learning. In AAAI, 2008.
>
> [3] Chan et al. Scalable Bayesian inverse reinforcement learning. In ICLR, 2021

---

> ### Author Response · Authors · 2022-08-09
> **A follow-up message to reviewer bSdL**
>
> The reviewer raised a few concerns mostly due to misunderstandings. They were clarified in the author response (our original paper is correct). Moreover, we developed a theorem to address the reviewer’s concern. We are wondering if you have any questions after taking a look at our response and the new theorem.
>
> The discussion period will end in about 12 hours. We authors will take on-call shifts to ensure we can answer your questions any time. Thank you!

---

### Official Review · Reviewer_iPjS · 2022-07-10

**Rating:** 6
**Confidence:** 2
**Soundness:** 3 good
**Presentation:** 3 good
**Contribution:** 3 good

**Summary:**

This work proposes a novel reward function for text generation models, which is derived from a pre-trained supervised model. Authors also suggest using the new reward function to utilize monolingual data for fine-tuning. This training framework aims to resolve two main weaknesses of text generation models, namely exposure bias and lack of parallel data. Empirical results on two different generation tasks show improvements upon the baselines.

**Questions:**

Comments and questions:
- line 12: You say "common training method" and give one citation from 2014. It does not sound convincing. At least you can cite Transformer model/Bert/BART
- While you speak about text generation training, do you mean the structure of the dataset or the input/output of the model by parallel dataset? Because the text generation model can be trained on monolingual data without any need for labels.
- You claim that large models intensify the problem of creating parallel datasets. But these large models are usually trained on **monolingual data**, and can be fine-tuned on the smaller task-specific dataset
- line 34: what objective do you use for training seq2seq?
- line154: you do not say what seq2seq model: is it Transformer? or T5?
- Table 1: Do I understand the training process correctly: T5 base pretrained --> train it on task-parallel data with teacher forcing --> to add additional parallel/non-parallel data using RL? Which of that does line 3 Table 1(a) (T5-Base) report?
- You cite results from "DialogBERT: Discourse-Aware Response Generation via Learning to Recover and Rank Utterances". Why can the numbers from your paper not be found in the source? Same for AdaLabel
- According "Exploring Diverse Expressions for Paraphrase Generation", the quora paraphrase test set contains 30k, you report 20k on test set. Why such difference?
- "Exploring Diverse Expressions for Paraphrase Generation" does not report iBLEU4, but you cite the number. What is this number?
- "Learning to Selectively Learn for Weakly-supervised Paraphrase Generation" does not report SBLEU, how did you get the number?

**UPD**: Questions are mostly answered by authors; Score is updated

**Limitations:**

Limitations are addressed in Section 5

**Strengths And Weaknesses:**

Strengths:
- Building an interesting connection between training with teacher-forcing objective and RL objective
- Significant improvements over the baseline (however, see weaknesses)

Weaknesses:
- The cited previous results in Table 1 need verification
- Sloppy citations
- Paper is hard to understand and needs clarification
- Thus: the argumentation in the paper is not convincing

**UPD**: Weaknesses are addressed by authors during discussion; Score is updated

---

> ### Author Response · Authors · 2022-08-01
> **Response to Reviewer iPjS (1/2)**
>
> We thank the reviewer for the comments.
>
> The reviewer gave an extradinatory score of 3, mainly concerning our experiments. For example, our baseline scores are worse than the original papers in the dialogue task (Weaknesses).
>
> This is because the original dialogue datasets are problematic: more than 20-30% samples are overlapping between training and test sets [1]. This very largely inflates model performance and makes comparison meaningless.
>
> We instead worked on the deduplicated dataset and quoted baseline results from [1]. The big gap, in fact, highlights the importance of working on correct datasets. This also demonstrates the scientific rigor in our experimentation.
>
> We understand that this issue was only recently realized by [1], and that the reviewer may not be aware of this. We sincerely hope the reviewer can read our response (and probably also the above paper), revisit our submission, and render a more convincing score.
>
> [1] Wen et al., An empirical study on the overlapping problem of open-domain dialogue datasets. In Proceedings of the Language Resources and Evaluation Conference, 2022.
>
> ---
>
> Detailed responses are as follows.
>
> > line 12: You say "common training method" and give one citation from 2014. It does not sound convincing. At least you can cite Transformer model/Bert/BART
>
> We cited the early paper because we made efforts to trace the origin of the “common training method”. We thank the reviewer for the suggestion and will include recent Transformer models.
>
> > While you speak about text generation training, do you mean the structure of the dataset or the input/output of the model by parallel dataset? Because the text generation model can be trained on monolingual data without any need for labels.
>
> In this paper, we consider conditional text generation (seq2seq, or encoder-decoder), such as dialogue generation and paraphrase generation. Text generation models pretrained on monolingual data typically also require parallel data for fine-tuning. This is exactly what we did: we took T5 and finetuned it for the task at hand.
>
> > You claim that large models intensify the problem of creating parallel datasets. But these large models are usually trained on monolingual data, and can be fine-tuned on the smaller task-specific dataset
>
> We agree that large language models make it feasible to fine-tune only on small datasets. However, this is not the whole story. In the machine translation (MT) domain, for example, WMT19 En-De dataset contains 38 million samples. State-of-the-art MT models [2] are generally trained/finetuned on the full dataset, which is by no means small.
>
>
>
> [2] Chen et al., Facebook AI’s WMT20 News Translation Task Submission. In Proceedings of the Conference on Machine Translation, 2020
>
> > line 34: what objective do you use for training seq2seq?
>
> We used the cross-entropy loss for training seq2seq. Line 34 is in the Introduction section. The formula of the objective (cross-entropy loss) is presented in Eq 4 (Section 2).
>
> > line154: you do not say what seq2seq model: is it Transformer? or T5?
>
> In Line 155, we stated that the model is initialized by the pre-trained T5.

---

> > ### Author Response · Authors · 2022-08-01
> > **Response to Reviewer iPjS (2/2)**
> >
> > > Table 1: Do I understand the training process correctly: T5 base pretrained --> train it on task-parallel data with teacher forcing --> to add additional parallel/non-parallel data using RL? Which of that does line 3 Table 1(a) (T5-Base) report?
> >
> > Yes, you are correct. The procedure was explained in Lines 154-157. Line 3 (Table 1a) reports the T5-Base model trained on the task-parallel data with teacher forcing.
> >
> > > Table 1 (a): You cite results from "DialogBERT: Discourse-Aware Response Generation via Learning to Recover and Rank Utterances". Why can the numbers from your paper not be found in the source? Same for AdaLabel.
> >
> > As said at the beginning of this response, the previous paper worked on an overlapping (thus wrong) dataset. We instead worked on deduplicated datasets [1], as we presume scientific research should follow the correct setting. We emphasized this in Line 177.
> >
> > > Table 1 (b): According "Exploring Diverse Expressions for Paraphrase Generation", the quora paraphrase test set contains 30k, you report 20k on test set. Why such difference?
> >
> > As mentioned in Lines 187-192, the paraphrase generation dataset is constructed from the QQP dataset and does not have a standard train/valid/test split. There are two common ways to split the dataset (see Quora-S and Quora-U settings in [3]). The 30K test split is common for the purely supervised setting, and a 20K test split is usually used in the unsupervised setting [4,5].
> >
> > We followed the second split, since our semi-supervised setting also incorporates the non-parallel dataset. All the semi-supervised competing methods adopt the same split and thus are directly comparable. This means that the comparison is fair between our method and semi-supervised baselines.
> >
> > [3] Ding et al., Learning to Selectively Learn for Weakly-supervised Paraphrase Generation. In Proceedings of the Conference on Empirical Methods in Natural Language Processing, 2021
> >
> > [4] Liu et al., Unsupervised paraphrasing by simulated annealing. In Proceedings of Association for Computational Linguistics, 2020
> >
> > [5] Li et al., Unsupervised Text Generation by Learning from Search, In Proceedings of Advances in Neural Information Processing Systems, 2020
> >
> > > Table 1 (b): "Exploring Diverse Expressions for Paraphrase Generation" does not report iBLEU4, but you cite the number. What is this number?
> >
> > As mentioned in Lines 193-197, the relationship of these metrics is iBLEU = (1-alpha) BLEU - alpha SBLEU. Thus, we are able to calculate iBLEU ourselves, given the reported BLEU and SLBEU. (alpha = 0.1 in our setting.)
> >
> > > Table 1 (b): "Learning to Selectively Learn for Weakly-supervised Paraphrase Generation" does not report SBLEU, how did you get the number?
> >
> > Again, this time SBLEU = [(1-alpha)BLEU - iBLEU] / alpha. (Here, alpha is not 0.)
> >
> > ---
> >
> > We thank the reviewer again for carefully checking the experimental setups. By walking through all these details together, we’re further confident that our experiments are correct and rigorous, convincingly showing the effectiveness of our approach. We sincerely hope the reviewer could reconsider the score. Thanks.

---

> > > ### Comment · Reviewer_iPjS · 2022-08-07
> > > **Reviewer comment after authors response; Update the score**
> > >
> > > Dear authors,
> > >
> > > I understand your frustration, and I carefully read your response.
> > >
> > > > In Line 155, we stated that the model is initialized by the pre-trained T5.
> > >
> > > You can initialize other models with pre-trained weights. So `model initialized by the pre-trained T5` is not necessarily equal to the pre-trained T5 model by itself.
> > >
> > > > [59] Wen et al., An empirical study on the overlapping problem of open-domain dialogue datasets. In Proceedings of the Language Resources and Evaluation Conference, 2022.
> > >
> > > In your paper, you specifically cited the original works in table 1 (a,b) and pointed out that `Numbers for models labeled
> > > with † are quoted from previous work`. That makes it extremely confusing. It would be best if you made it clear what you are citing and what you compute whenever possible.
> > > Again, even if I look at the numbers in the paper, it is hard to find where the numbers came from, even if I am taking [59] as a reference.
> > > And if you used a cleaned dataset, have you used single-term or multi-term? Hard to guess given split numbers.
> > >
> > > I am more than willing to raise my score to **5**, but I kindly ask you to describe the setup and numbers you report carefully.
> > >
> > > Best,
> > > Reviewer iPjS

---

> > > > ### Author Response · Authors · 2022-08-08
> > > > **Thank you for the response**
> > > >
> > > > Thank you so much for your response. During the author response and discussion periods, we have been mainly focusing on conducting additional experiments and developing a new theorem according to all reviewers’ suggestions. We’re continuously improving our manuscript and will especially clarify the T5 pertained model and the source of quoted numbers in the revision. Thanks again.

---

> > > > > ### Comment · Reviewer_iPjS · 2022-08-08
> > > > > **Update**
> > > > >
> > > > > Dear authors,
> > > > >
> > > > > thanks for your effort and refinements. I believe paper presentation is tremendously important and it is upsetting when good work is not recognized because of that. I've updated my score again, but since I can not see your final version, I settle on 6.
> > > > >
> > > > > All the best,
> > > > > Reviewer iPjS

---

### Official Review · Reviewer_s2sH · 2022-07-10

**Rating:** 7
**Confidence:** 4
**Soundness:** 2 fair
**Presentation:** 3 good
**Contribution:** 2 fair

**Summary:**

This paper presents an approach for training a text-generation model by transforming an existing conditioned language model into a reward function. The paper suggests that the pre-softmax logits of a pre-trained sequence-to-sequence language model can be used to compute a per-token reward for new input sequence data. The paper also proposes an algorithm for training a model with this reward function, including off-policy exploration. The experiments compare the proposed approach against several other methods for training with additional non-parallel data (e.g., self-training, and R-regression, which uses a reward function derived from BLEU score. Compared to existing approaches which use single-sentence reward (e.g., BLEU score), the proposed approach provides a denser reward which results in moderate improvement over the compared systems.

**Questions:**

* What is the intuition behind the derived reward function in Eq 7? It looks kind of like an advantage function.
* What happens if r(s, a) < 0, if this ever happens (it seems like it should)? Does this cause problems with training, as this means Alg 1 is directly minimizing log \pi?
* Would training with teacher forcing basically be \pi_b \prop f_w, with k = \inf?
* How is \pi_\phi initiated? With f_w or just the original T5 base parameters?
* Is the training for self-training using teacher forcing, just with sequences derived via inference from unpaired data?
* How do you choose to stop training?
* Eq 7 -- f_w returns unbounded logits, right? Does this cause issues for training?
* Did you experiment with a discounting factor in Algorithm 1 that attenuates return backwards in time? If f_w is usually positive, this seems like it would give much higher rewards to actions at the beginning of a sequence rather than at the end.
* Did you experiment with an entropy term to avoid collapse (similar to what you are seeing with k=1)?

**Limitations:**

I'd be interested in hearing a bit about computational efficiency for training and how the different methods compare. Self-training seems more efficient as it doesn't require sampling/inference at training time, except for a fixed number of examples. I believe R-regression requires running inference on a separate model, and obviously the proposed approach does as well, but it seems twice as many times, as in Eq 7 f_w is computed both for s and s + [a].

**Strengths And Weaknesses:**

Strengths:

To the best of my knowledge, I haven't seen any other work that trains a model with RL using rewards derived from another language model. It is a clever approach. Experiments were done on two different text generation datasets.

---

Weaknesses:

The improvements in performance reported in Table 1 seem quite small, except for BLEU2. Are they significant?

There are some experiments I would have liked to see:
* The ability to compute the proposed reward function is general, but it still requires a domain-specific f_w to compute. Would have been nice to see experiments on deriving reward functions from other f_w besides in-domain, e.g., rewards from a general pre-trained language model.
* How much does the quality of f_w matter? It would have been interesting to see performance relative to the number of training examples used to train f_w. How little parallel data do you need?
* It would be nice to see learning curves / reward curves comparing the proposed approach with other methods. How many samples and gradient updates do you need for the proposed approach vs. self-training, R-regression, etc?

And some evaluation I would have liked to see:
* Some deeper analysis of model outputs after different training methods. E.g., more than a one or two pieces of generated text per approach.
* Evaluation with a more recent text generation metric, e.g. BERTScore or similar

Some claims / phrasing are a bit confusing:
* Teacher-forcing isn't necessarily the only way to train a text generation model; masked training objectives are also effective and popular these days. Also, you don't need parallel data to use teacher-forcing based training objectives.
* I'd also be interested to see non-conditional text generation with this proposed method. It doesn't seem like anything in the proposed method requires the task to be a sequence-to-sequence task (maybe this is what the authors are suggesting for future work with the reference to GANs in L153?).

Some missing related work:
* Kreutzer et al. 2016 / 2017 / 2018 use BLEU score for training text generation

---

Some minor suggestions (e.g. readability):

* In 2.1, I'd suggest moving lines 61 and 64 to closer to the top[ (e.g., after L56). I also am not sure what Eq 2 adds, besides it being standard in RL papers.
* What does it mean for a reward function to be "naturally defined" in the context of text-generation; why would the suggested reward function be "natural"? (L65)
* L100 -- policy q should be \pi_q, right?
* The discussion in the beginning of 2.3 is a bit confusing; the term "behavior policy" is unfamiliar to me ("inference policy" seems more natural)
* The claim that any behavior policy can generate any trajectory seems broad to me. What if \pi_b = \pi_\phi?
* Nitpick on L130 -- the policy starts to generate fixed trajectories because the entropy is collapsing, not because it's becoming more optimal. Also in L254, this seems to be the case as well.
* I was relatively confused about what was being plotted in Fig 1 -- what's the x-axis?

---

> ### Author Response · Authors · 2022-08-01
> **Response to Reviewer s2sH (1/4)**
>
> We thank the reviewer for recognizing the novelty of our work and saying it is “a clever approach.”
>
> ---
>
> > Weakness: “The improvements in performance reported in Table 1 seem quite small, except for BLEU2. Are they significant?”
>
> We found the relative improvement is decent, although the absolute difference appears small. We know that dialogue utterances are diverse, and thus it is generally hard to achieve a high BLEU score. We achieve a relative improvement of 3%. This is analogous to 1-point improvement from a BLEU score of 30 in machine translation, which is usually considered significant. Moreover, we conducted experiments on two datasets with different metrics. Results are generally consistent.
>
> We’re conducting a statistical analysis and will report the result when it’s ready (hopefully during the discussion period).
>
> > Weakness: “There are some experiments I would have liked to see:”
> >> Point 1: The ability to compute the proposed reward function is general, but it still requires a domain-specific f_w to compute. Would have been nice to see experiments on deriving reward functions from other f_w besides in-domain, e.g., rewards from a general pre-trained language model.
>
> Thanks for the insightful suggestion! In fact, we are thinking of extracting rewards from large language models in a zero-shot manner using prompts as future.
>
>
> >> Point 2: How much does the quality of f_w matter? It would have been interesting to see performance relative to the number of training examples used to train f_w. How little parallel data do you need?
>
> Thanks for the suggestion. We conducted the experiment with preliminary results as follows
>
> BLEU2 on DailyDialog
>
> Model            |      10%  |  30%  | 100% of training data
> ---| --- | --- |---
> Self-Training | 8.38  |  8.39  |  9.10
> R-Regression|  9.24  |  9.50  |10.34
> Our approach|   10.27  | 10.48  |  11.02
>
> The results show that a high-quality $f_w$ definitely leads to better performance. But our model is still robust with $f_w$. We’re happy to include the results as a plot in the revision.
>
> >> Point 3: It would be nice to see learning curves / reward curves comparing the proposed approach with other methods. How many samples and gradient updates do you need for the proposed approach vs. self-training, R-regression, etc?
>
> We have reported the learning curve of our model in Figure 2. We see that at the beginning our curves grow gradually. After updating the policy we see a large improvement, and finally the performance remains stable. We expect the learning curve of self-training grows more smoothly (because it doesn’t have periodic updates) but is lower than our performance in general.
>
> >Weakness: “And some evaluation I would have liked to see”
> >> Point 1: Some deeper analysis of model outputs after different training methods. E.g., more than one or two pieces of generated text per approach.
>
> We have shown certain interesting analyses in the paper. For example, paraphrasing models tend to generate output sentences similar to the input. Self-training learns from its own output, thus achieving a high Self-BLEU score (i.e., BLEU against input), which is undesired for paraphrase generation. By contrast, our RL method is able to explore the entire sentence space; it improves the main metric iBLEU without inflating Self-BLEU.
>
> If the reviewer provides more concrete suggestions, we will be very grateful to the reviewer and willing to conduct additional analyses.
>
> >> Point 2: Evaluation with a more recent text generation metric, e.g. BERTScore or similar
>
> We chose standard metrics from previous work for the evaluation of competing methods. Using BERTScore will make it difficult to compare with certain baseline models in previous papers. Lacking BERTScore doesn’t affect our conclusion as our results are consistent in different tasks in terms of different metrics. We’ll consider BERTScore for other applications in future work. Thanks for the suggestion.

---

> > ### Author Response · Authors · 2022-08-01
> > **Response to Reviewer s2sH (2/4)**
> >
> > > Weakness: Some claims / phrasing are a bit confusing:
> > >> Point 1: Teacher-forcing isn't necessarily the only way to train a text generation model; masked training objectives are also effective and popular these days.
> >
> > This is correct. However, masked training objectives are usually for non-autoregressive (NAR) text generation. Most language models (e.g., T5, GPT-3) are still trained with the teacher-forcing objective. Extending our approach to the NAR case is an interesting idea. We hope this paper can open a wide future work direction in terms of different frameworks, settings, and applications.
> >
> > >> Also, you don't need parallel data to use teacher-forcing based training objectives.
> >
> > We agree. We used parallel data because we mainly considered conditional text generation (see below for more discussion). Nevertheless, our approach is compatible with the non-parallel teacher-forcing as well. As mentioned, we plan to induce the reward by prompting pretrained language models in a zero-shot manner.
> >
> > >> Point 2: I'd also be interested to see non-conditional text generation with this proposed method. It doesn't seem like anything in the proposed method requires the task to be a sequence-to-sequence task (maybe this is what the authors are suggesting for future work with the reference to GANs in L153?).
> >
> > Thank you for the suggestion. Yes, we mainly consider conditional text generation because the tasks are clearly defined and the evaluation metrics are well established. This brings a more rigorous comparison of different methods. Non-conditional text generation could be considered in future work.
> >
> > >> Weakness: Some missing related work:  Kreutzer et al. 2016 / 2017 / 2018 use BLEU score for training text generation
> >
> > Thanks. We already discussed and compared RL with BLEU as the reward. We will include this line of work for discussion in the revision.
> >
> >
> > > Some minor suggestions:
> > >> Point 1: In 2.1, I'd suggest moving lines 61 and 64 to closer to the top[ (e.g., after L56). I also am not sure what Eq 2 adds, besides it being standard in RL papers.
> >
> > We included Eq 2 because it is used for deriving the reward (Line 114).
> >
> > >> Point 2:  What does it mean for a reward function to be "naturally defined" in the context of text-generation; why would the suggested reward function be "natural"? (L65)
> >
> > By “natural,” we mean that our reward does not rely on external heuristics (e.g., text length, BLEU scores). The main contribution of this paper is to propose the principled and “natural” approach to derive the reward function for text generation.
> >
> > >> Point 3: L100 -- policy q should be \pi_q, right?
> >
> > Thanks for pointing out the typo. It should be “value function $q$”.
> >
> > >> Point 4: The discussion in the beginning of 2.3 is a bit confusing; the term "behavior policy" is unfamiliar to me ("inference policy" seems more natural)
> >
> > The “behavior policy” means the one used to sample sentences during training. (This may be different from the policy used in inference.) We used the term because it’s conventional in the RL literature. We’re happy to clarify in the revision.
> >
> > >> Point 5: The claim that any behavior policy can generate any trajectory seems broad to me. What if \pi_b = \pi_\phi?
> >
> > Thanks for pointing this out. Here, we didn’t mean it is theoretically guaranteed, but it’s a common phenomenon in practice. We revised the statement as:
> >
> > In practice, off-policy REINFORCE ($\pi_\varphi \ne \pi_b$) is more exploratory than the on-policy one ($\pi_\varphi=\pi_b$), because the model policy $\pi_\varphi$ would become more concentrated during optimization and does not explore much, whereas $\pi_b$ is typically chosen to cover more trajectories.
> >
> > >> Point 6: Nitpick on L130 -- the policy starts to generate fixed trajectories because the entropy is collapsing, not because it's becoming more optimal. Also in L254, this seems to be the case as well.
> >
> > Thanks for pointing this out. We’ll clarify this (as shown above).
> >
> > >> Point 7: I was relatively confused about what was being plotted in Fig 1 -- what's the x-axis?
> >
> > The x-axis is the scale of $\hat{q}_t^r$, i.e., the estimated q-value functions along sampled sentences, mentioned in the caption.

---

> > > ### Author Response · Authors · 2022-08-01
> > > **Response to Reviewer s2sH (3/4)**
> > >
> > > > Q1: What is the intuition behind the derived reward function in Eq 7? It looks kind of like an advantage function.
> > >
> > > The intuition is that, given a q-value function (which is derived from the policy by Assumption 1), we will be able to derive a reward function as Eq 7 by rearranging the Bellman optimality equation (Eq 2).
> > >
> > > In RL, the (optimal) advantage function is defined as
> > > $A(s,a) = q(s, a) - v(s) = q(s,a) - \max_{a’} q({\color{red}s}, a’)$
> > >
> > > But our derived reward is
> > > $r(s,a) = q(s,a) - \max_{a’} q({\color{red}s+[a]}, a’)$
> > >
> > > The difference is highlighted in red. In other words, there is a one-step shift in the second term.
> > >
> > > More importantly, the advantage is defined to have a relative comparison among different q-values given a state (e.g., for actor-critic training), where the reward is typically assumed to be well-defined and given.
> > >
> > > We instead derive the reward from a given q value function. Therefore, we believe they are not related although they appear similar.
> > >
> > > > Q2: What happens if r(s, a) < 0, if this ever happens (it seems like it should)? Does this cause problems with training, as this means Alg 1 is directly minimizing log \pi?
> > >
> > > $r(s,a)<0$ is possible and does not cause trouble. This is especially allowed in policy gradient, as $r(s,a)<0$ means the actions are (generally) bad, so we minimize $\log\pi$ in Alg 1 to avoid them. Thus, no special treatment is needed.
> > >
> > > > Q3: Would training with teacher forcing basically be \pi_b \prop f_w, with k = \inf?
> > >
> > > If we do self-training (which also involves teacher forcing based on self-generated samples), yes, we do have $\pi_b \propto \exp f_w$ with $k = \infty$, as $\pi_b$ is never updated. However, there are still differences, as self-training learns by cross-entropy loss but our method learns by the induced reward.
> > >
> > > If we consider supervised self-training with parallel data, the answer is no, because we do not have the notion of $\pi_b$.
> > >
> > >
> > > > Q4: How is \pi_\phi initiated? With f_w or just the original T5 base parameters?
> > >
> > > It is initialized with $f_w$, whose parameters are initialized by T5 but fine-tuned on (small) parallel data.
> > >
> > >
> > > > Q5: Is the training for self-training using teacher forcing, just with sequences derived via inference from unpaired data?
> > >
> > > Yes. Old-day self-training classification uses its output category as pseudo-groundtruth and learns it by cross-entropy loss. Thus, a straightforward extension to generation tasks is to treat self-generated text as pseudo-groundtruth too and learn it by teacher forcing.
> > >
> > > > Q6: How do you choose to stop training?
> > >
> > > We set the maximum training step to be 10K and picked the best model based on validation scores. We’ll provide more details.
> > >
> > > > Q7: Eq 7 -- f_w returns unbounded logits, right? Does this cause issues for training?
> > >
> > > Yes, it is unbounded. As mentioned in Appendix A, we adopt the common strategy [1] to clip the unbounded rewards to [-1, 1]. However, we do not believe this is an important issue as most logits are within a reasonable range.
> > >
> > > > Q8: Did you experiment with a discounting factor in Algorithm 1 that attenuates return backwards in time? If f_w is usually positive, this seems like it would give much higher rewards to actions at the beginning of a sequence rather than at the end.
> > >
> > > No, we did not use a discounting factor because it is not well motivated to discount the reward for text generation. Since all sequences (texts) are finite, the discounting factor is not mandatory.
> > >
> > > $f_w$ may not always be positive because it’s simply certain neural logits, i.e., the values after linear projection but before softmax. Thus, it does not emphasize the beginning of a sequence more. Recall the negative rewards are compatible with policy gradient (see Q2).
> > >
> > > > Q9: Did you experiment with an entropy term to avoid collapse (similar to what you are seeing with k=1)?
> > >
> > > Yes. As mentioned in Appendix A, we use label smoothing (or log-barrier regularization) for all experiments, but it does not prevent the policy from collapsing when $k=1$.
> > >
> > > > Limitations:  I'd be interested in hearing a bit about computational efficiency for training and how the different methods compare. Self-training seems more efficient as it doesn't require sampling/inference at training time, except for a fixed number of examples. I believe R-regression requires running inference on a separate model, and obviously the proposed approach does as well, but it seems twice as many times, as in Eq 7 f_w is computed both for s and s + [a].
> > >
> > > Interesting thoughts! But our computation is not twice as many as R-Regression, because computing $f_w(s+[a])$ will be reused as $f_w(s)$ in the next step. In other words, we only need one pass of Transformer and the efficiency is identical to R-Regression
> > >
> > >
> > > [1]  Mnih et al. Human-level control through deep reinforcement learning. Nature, 2015

---

> > > > ### Author Response · Authors · 2022-08-01
> > > > **Response to Reviewer s2sH (4/4)**
> > > >
> > > > ---
> > > >
> > > > Summary
> > > >
> > > > We thank the reviewer again for detailed comments. Since the reviewer highly recognizes our novelty (as he/she hasn’t “seen any other work” doing similar things), the reviewer also suggested a number of extensions, new use cases, and new applications.
> > > >
> > > > We would like to point out that the main contribution of this paper is to lay the foundation of reward induction, instead of being a solely empirical paper. As we have already overflowed into the appendix, we would not be able to include all these things in a single paper. We hope our work can open a new research direction for ourselves as well as peer researchers in the community.

---

> > > > > ### Comment · Reviewer_s2sH · 2022-08-09
> > > > > **Thank you**
> > > > >
> > > > > Thank you for the detailed response! I will update my score to 7.

---

> ### Author Response · Authors · 2022-08-07
> **Significance test**
>
> We have conducted the significance test to compare our model and R-Regression. We implemented the script ourselves because the test for iBLEU is not supported by existing toolkits. Our implementation of the significance test is based on paired bootstrap resampling [1]. We set the bootstrap number as 1000 and the sampling ratio as 1.0. The results of BLEU2 and BLEU4 match the scores generated by the existing library [2]. The results of p-values are as follows:
>
> DailyDialog \
> BLEU2: <0.01, BLEU4: <0.05
>
> Quora \
> iBLEU4, BLEU4: both <0.01
>
> Since both test sets are large enough (~6K and ~20K), we believe the comparisons are statistically significant. We hope this experiment addresses the concern of the significance of our improvements.
>
> [1] Koehn. Statistical Significance Tests for Machine Translation Evaluation. EMNLP 2004.\
> [2] Koehn et al. Moses: Open Source Toolkit for Statistical Machine Translation. ACL 2007

---

> ### Author Response · Authors · 2022-08-09
> **A follow-up message to reviewer s2sH**
>
> We thank the reviewer for caring about our paper in great detail. We’ve conducted several additional experiments according to the suggestions provided, where results consistently suggest the effectiveness of our approach.
>
> Should you have any last-minute questions, we would be able to answer them any time before the discussion period ends. Thanks!

---

### Official Review · Reviewer_BSmj · 2022-07-11

**Rating:** 7
**Confidence:** 3
**Soundness:** 3 good
**Presentation:** 3 good
**Contribution:** 4 excellent

**Summary:**

The paper draws a connection between teacher-forcing and inverse reinforcement learning. The authors derives theoretically from the teacher-forcing objective to an equivalent reward function (i.e., a model trained with teacher-forcing recovers a reward function). With this insight, the authors leverage this learned reward function to train on non-parallel data. The reward can be applied step-wise, therefore mitigates the common instability issues when training seq2seq models.

**Questions:**

- In equation 7, the second term requires computing a $\max_{a' \in \mathcal{A}} f_w(s + [a], a')$. Does this mean that at each generation step the model needs to run a forward path for every action, i.e., word in the vocabulary? If so it might render the method slow and hard to scale.
- The Periodically Synchronized Behavior Policy is described as a contribution but to my knowledge similar issues (discrepancy between the behavior policy and the model policy) are long-standing and has been discussed as a simple fix such as in PPO (Schulman et al., 2017) or other off-line RL methods (update after K steps). Is there any main difference that I missed here?

**Limitations:**

The authors appropriately addressed potential limitations in a standalone section.

**Strengths And Weaknesses:**

Strengths
- The idea that links teacher-forcing to its inherently learned reward function is, to my knowledge, novel and the theoretical connection has not been explicitly drawn before
- The paper is easy to follow and the methods are well motivated
- The proposed method is simple and easy for the community to extend and build research upon
- The experimental results and the analysis are illuminative

Weaknesses phrased as questions below (if addressed or answered, I'm happy to change the score).

---

> ### Author Response · Authors · 2022-08-01
> **Response to Reviewer BSmj**
>
> Thanks for the insightful review and your support! The reviewer fully recognizes the novelty, theoretical value, and importance of our work.
>
> ---
>
> > Q1: In equation 7, the second term requires computing $\max_{a′ \in A} f_w(s+[a],a′)$. Does this mean that at each generation step the model needs to run a forward path for every action, i.e., word in the vocabulary? If so it might render the method slow and hard to scale.
>
> A: Thank you for the question. No, we do not need multiple forward passes for a sample.
>
> Our RL training assumes a trajectory is sampled, so $[a]$ is already determined when we calculate Eq (7). Moreover, $f_w(s+[a], \cdot)$ is implemented as the logit of a softmax layer; computing $\max$ over $a’$ is simply computing maximum of softmax logits. Therefore, only one forward pass is needed for a sample.
>
> > Q2: The Periodically Synchronized Behavior Policy is described as a contribution but to my knowledge similar issues (discrepancy between the behavior policy and the model policy) are long-standing and has been discussed as a simple fix such as in PPO (Schulman et al., 2017) or other off-line RL methods (update after K steps). Is there any main difference that I missed here?
>
> A: Thanks for the comment. Yes, our general idea is similar to PPO, but here we found such a simple fix is also effective to off-policy REINFORCE, which is much simpler than PPO. In the revision, we’d clarify our simple yet effective finding, and include PPO and other related work for discussion. In general, our main contribution is still the connection between teaching-forced seq2seq training and IRL.
>
> ---
>
> We thank the reviewer again for the review. We hope our response has addressed both questions, and are especially grateful to the reviewer’s willingness to change the score. Should there be further questions, please do not hesitate to let us know during the discussion period.

---

> ### Author Response · Authors · 2022-08-09
> **A follow-up message to reviewer BSmj**
>
> Thank you again for recognizing the contribution of our work and your willingness to change the score.
>
> We have addressed both of your questions (computational efficiency and discussion with PPO) in the response and will incorporate them into the revision. Please feel free to discuss if you have any other questions unanswered. Thanks!

---

### Author Response · Authors · 2022-08-07
**A follow-up message regarding the rebuttal**

Dear reviewers:

We thank your efforts in reviewing our paper and providing insightful suggestions. We have addressed all the concerns raised and will incorporate them in the revision.

With the reviewer-author discussion deadline approaching, we would highly appreciate it if reviewers could take a look at our responses. Should there be any further questions, please do not hesitate to ask us.

Thank you very much, \
-Authors

---

### Author Response · Authors · 2022-08-09
**Thanks for the discussion**

We thank all reviewers for the insightful discussion and suggestions. We really appreciate your time and efforts!

---

### Meta-Review · Area_Chair_eutQ · 2022-08-26

**Recommendation:** Accept
**Confidence:** Certain

**Metareview:**

This paper proposes a method to design a reward function from a pre-trained language model, and uses it to train a text generation model using reinforcement learning. The approach is novel and the paper presents both theoretical derivations (drawing connections between maxent IRL and the supervised teacher forcing loss) and empirical results to demonstrate the effectiveness of the proposed algorithm, offering new insights for a challenging problem.

**Award:**

No

---

### Decision · Program_Chairs · 2022-09-14

Accept